Deteriorated sleep quality and associate factors in patients with type 2 diabetes mellitus complicated with diabetic peripheral neuropathy

Fu Lin 1 2
Zhong Liping 1
Liao Xin 3
Wang Lingrui 4
Wang Youyi 1
Shi Xiuquan xqshi@zmu.edu.cn 1 2
Zhou Yanna zhouyanna@zmu.edu.cn 1 2
1 Department of Epidemiology and Health Statistics, Zunyi Medical University , Zunyi , China
2 Key Laboratory of Maternal & Child Health and Exposure Science of Guizhou Higher Education Institutes , Zunyi , China
3 Endocrinology Department, Affiliated Hospital of Zunyi Medical University , Zunyi , China
4 Endocrinology Department, the Second Affiliated Hospital of Zunyi Medical University , Zunyi , China
Foti Daniela
Electronic publication date: 2024 Jan 22
Publication date: 2024
Volume: 12
Electronic Location ID: e16789
Received 2023 Jul 14; Accepted 2023 Dec 21
Copyright: ©2024 Fu et al.
Copyright year: 2024
Copyright holder: Fu et al.
License: This is an open access article distributed under the terms of the Creative Commons Attribution License, which permits unrestricted use, distribution, reproduction and adaptation in any medium and for any purpose provided that it is properly attributed. For attribution, the original author(s), title, publication source (PeerJ) and either DOI or URL of the article must be cited.
License URL: https://creativecommons.org/licenses/by/4.0/

Keywords: Type 2 diabetes mellitus, Diabetic peripheral neuropathy, Sleep quality, Influence factors

Funding: The Innovation training program for college students of Zunyi Medical University ZYDC202202103 The Science & Technology Program of Guizhou Province QKHPTRC-CXTD [2022]014 The Master’s Program Construction Funds of Zunyi Medical University SSDJS201818 This work was supported by the Innovation training program for college students of Zunyi Medical University (NO: ZYDC202202103), the Science & Technology Program of Guizhou Province (Grant No. QKHPTRC-CXTD [2022]014), and the Master’s Program Construction Funds of Zunyi Medical University (Grant No. SSDJS201818). The funders had no role in study design, data collection and analysis, decision to publish, or preparation of the manuscript.

==============================
Objectives

To understand the sleep quality and its influencing factors in patients with type 2 diabetes mellitus (T2DM) who suffered diabetic peripheral neuropathy (DPN), and provide evidence for clinicians to carry out comprehensive intervention measures to improve the sleep quality of patients.

Methods

Patients who were admitted to the Endocrinology Department of Affiliated Hospital of Zunyi Medical University were recruited from May to December 2022, and the investigation were conducted by face-to-face interview. The questionnaires included PSQI questionnaire and influencing factors, such as lifestyle and health status.

Results

Among the 193 patients, 40.4% of the patients never took physical examination, 56.5% of the patients had duration of illness greater than 5 years, 61.7% of the patients had had an operation, 10.4% of the patients had bad dietary status, and 55.4% of the patients had physical pain. In addition, the PSQI general score was 8.34 ± 3.98, the occurrence rate of poor sleep quality (PSQI ≥ 8) was 54.4%, and the results showed that sleep quality of the physical pain group was worse than the no pain group. Moreover, the results of multivariate analysis revealed that the factors affecting sleep quality were lower frequency of exercise, bad dietary status, lower frequency of physical examination, longer duration of illness, and smoking, and the OR and 95% CI were [1.40, 1.04∼1.89], [3.42, 1.86∼6.29], [1.49, 1.01∼2.20], [1.78, 1.09∼2.92], [2.38, 1.17∼4.88], respectively.

Conclusion

Patients with DPN have higher risk of poor sleep quality. Moreover, there were many risk factors associated with poor sleep quality, clinicians and health policymakers should timely detect and effectively intervene in these factors to improve the sleep quality, which is important to enhance the quality of life of T2DM patients complicated with DPN.

Introduction

Type 2 diabetes mellitus (T2DM) is a metabolic disorder based on insulin resistance and pancreatic β-cell dysfunction, which is a clinically typical chronic disease (Society, 2021). China has one of the highest prevalence of diabetes mellitus in the world, and according to the World Health Organization (WHO) in 2016 (Yang et al., 2016), there are about 110 million people with diabetes mellitus in China, and this figure is expected to reach 150.7 million by 2040. A number of lifestyle behaviors, including bad dietary status, lack of physical activity, and sleep problems have contributed to the surge in T2DM patients (Cabrera-Mino et al., 2021).

Sleep is vital for maintaining health and social adaptability in humans. Healthy sleep behaviors are considered one of the key lifestyle components in the treatment of T2DM (Committee ADAPP, 2021). Sleeping problems are common in T2DM and cause disturbances in sleep quality and duration. Research has shown that insomnia, sleep apnea syndrome and other types of sleeping problems are strongly associated with the onset and progression of T2DM (Zhang et al., 2019). These sleep problems may lead to chronic inflammatory responses, neuroendocrine disruption and weight gain, which can increase the risk of developing diabetes (Fallahi et al., 2019; Zuraikat et al., 2020).

Among the many complications of diabetes, diabetic peripheral neuropathy (DPN) is one of the most common and early manifesting complications, which is a symptom of peripheral nerve dysfunction in diabetic patients when other causes are excluded (Gylfadottir et al., 2020; Iqbal et al., 2018). Previous studies have demonstrated a significantly higher incidence of complications in T2DM compared to non-T2DM individuals. (Amutha et al., 2017; Pan et al., 2018). A national retrospective study of diabetic inpatients from 1991-2000 showed that the prevalence of DPN was 60.3%, of which T2DM accounted for 61.8% (Society & Association, 2002). Pain, abnormal sensation and numbness in the hands and feet associated with DPN not only affect patients’ emotion and social function seriously, but also lead to a decline in sleep quality (Kioskli et al., 2019). What’s more, decreased sleep quality can exacerbate glucose metabolism disorders, lead to decreased insulin sensitivity and aggravate insulin resistance, which have a serious impact on the plasma glycemic control and quality of life of T2DM (Dong et al., 2020).

To our knowledge, most studies have focused on the relationship between sleep duration and plasma glycemic control in T2DM, as well as the influencing factors of sleep quality in T2DM patients. However, few studies have been conducted on the sleep status and influencing factors of T2DM suffering from complications. In addition, related studies have shown that 47.1% of patients with T2DM have poor sleep quality, and this percentage may be higher among those with DPN (Zhu, Quinn & Fritschi, 2018; Karmilayanti Goysal et al., 2021). This study analyzed the sleep quality and its influencing factors in patients with T2DM who suffered DPN and provided evidence for clinical workers to carry out comprehensive intervention measures to improve the sleep quality of patients.

Material and Methods

Study design

A cross-sectional survey was conducted between May to December 2022 in the Endocrinology Department of Affiliated Hospital of Zunyi Medical University.

The following inclusion criteria were used: (1) patients who met the World Health Organization’s diagnostic criteria for T2DM; (2) patients who had been diagnosed with DPN by region-current perception threshold (R-CPT); (3) patients with good cognitive function and can communicate normally. The following exclusion criteria were used: (1) patients with type 1 diabetes mellitus or other types of diabetes; (2) patients with acute complications of diabetes (diabetic ketoacidosis, hyperglycemic hyperosmolar state), acute cardiac insufficiency, severe liver and kidney abnormalities and malignant tumors; (3) patients with mental illness and severe hearing impairment who cannot cooperate to complete the questionnaire.

Questionnaire

The questionnaire consisted of three parts, including basic demographic data, the influence factors questionnaire, and the Pittsburgh Sleep Quality Index (PSQI) questionnaire.

The first part included the basic demographic data of age, sex, occupation, body mass index (BMI) and marital status. BMI were calculated based on the height and weight recorded in the patients’ admission medical records.

The second part was the influence factors questionnaire. By reviewing the relevant literature, we hold a point on that the questionnaire should contain influence factors as follows, such as smoking habits, drinking habits, history of hypertension, history of operation, physical pain, duration of illness, dietary status, frequency of exercise, frequency of physical examination, psychological burden, life pressure. Additionally, preliminary investigation results indicated an acceptable level of reliability and validity for the second part of the questionnaire, with a Cronbach’s alpha coefficient of 0.736 and a KMO coefficient of 0.754. In this study, smoking represented patients who were currently smoking or had smoked for more than a month in the past but had currently quit. The same definition applied to drinking habits. Operation was defined as the operation of grade three or above in the Operation Grading Management Measures for Medical Institutions issued by the National Health Commission of the People’s Republic of China within 10 years (China NHCotPsRo, 2022). History of hypertension was self-reported by patients, but hypertension was not categorized and graded. The physical pain came from the patients’ self-report based on their own pain conditions. Dietary status was reported by the patients based on their appetite in the last month, and good dietary status means having a healthy appetite and consuming an adequate amount of food. General dietary status means having a moderate appetite and consuming a moderate amount of food. Bad dietary status means having a poor appetite and consuming a significantly reduced amount of food. Exercise was defined as 30 minutes or more of moderate-intensity exercise, including brisk walking, jogging, stair climbing, etc. For the assessment of exercise frequency, we divided it into “≥5 times/week” “3-4 times/week” “1-2 times/week” and “hardly”. These classifications were based on the frequency of exercise reported by the patients. The frequency of physical examination referred to the number of times that patients went to medical and health institutions for health examination. Psychological burden meant psychological stress and burden due to having diabetes and related complications. Life pressure was defined as high levels of stress originating from work and life in general.

The third part was the PSQI questionnaire which is the most common measure of sleep quality. The questionnaire was developed by Buysse et al. (1989) in 1989 to evaluate the quality of sleep in the last one month. The Chinese version of the PSQI questionnaire has reported good internal consistency and test-retest reliability (Ho et al., 2021). And this questionnaire has been widely used to assess sleep quality currently (Liu et al., 2020; Zhang et al., 2023). It contains seven component scores: subjective sleep quality, sleep latency, sleep duration, habitual sleep efficiency, sleep disturbances, use of sleep medications, and daytime dysfunction. These scores are summed to yield a total PSQI score with a range of 0–21 and higher scores indicate poorer sleep quality. In our study, a PSQI score of <8 and ≥ 8 were defined as good sleep quality and poor sleep quality, respectively, based on previous research (Barakat et al., 2019).

Study sample

The sample size required for the study was estimated by the sample size calculation formula of the cross-sectional study. n =Z2 ×(P ×(1-P))/d2. A previous study showed that the prevalence of poor sleep quality among painless DPN patients was 64.5% (Choi et al., 2021). So, we set p = 0.645, the precision (two-sided) was 0.16, and the confidence level was 0.95, a sample size of 148 participants could be calculated by PASS 15.0 (NCSS, LLC, Kaysville, Utah, USA). Considering the 20% non-response rate, the minimum required sample size was 185 participants.

A total of 200 eligible T2DM patients were invited into the study. Among them, seven patients refused to participate in the survey. Finally, 193 individuals with T2DM complicated by DPN who met the criteria were enrolled in the study. The response rate was 96.5%. The study involving human patients was reviewed and approved by the Medical Research Ethics Committee of Affiliated Hospital of Zunyi Medical University (KLLY-2021-192). The patients provided their written informed consent to participate in this study. The survey was conducted using a uniform questionnaire and questioning methods, and the investigation was conducted with face-to-face inquiry and the results were recorded faithfully.

Statistical analysis

Statistical analyses were performed using SPSS 18.0 (IBM Corp., Armonk, NY, USA). Mean and standard deviation (x¯±s) were used to describe normally distributed measurement data, while the differences between groups were assessed using two-independent samples t-test. For abnormally distributed measurement data, the average rank was used for description, and the Mann–Whitney U test was used to test the differences between two groups. Count data was presented as rates or percentages. Logistic regression analysis was employed to assess the impact of variables on sleep quality. The significance level was set at α = 0.05. The logistic regression model was visualized in the form of a nomogram using the rms package (https://CRAN.R-project.org/package=rms) in R (R Core Team, Vienna, Austria) software.

Result

General profile of patients

The average age of the 193 patients was (56.12 ±11.59) years, with the oldest was 89 years old and the youngest was 19 years old. Among them, 60.1% (116/193) were male, 39.9% (77/193) were female, 39.9% (77/193) smoked, 36.8% (71/193) drank, 43.5% (84/193) hardly exercised, 40.4% (78/193) never took physical examination, 56.5% (109/193) had a duration of illness greater than 5 years, 61.7% (119/193) had had an operation, 10.4% (20/193) had bad dietary status, 55.4% (107/193) had physical pain, and 16.6% (32/193) had greater psychological burden. According to the guideline for the prevention and treatment of type 2 diabetes mellitus in China (2020 edition) (Society, 2021), the rates of meeting the standard were as follows: hemoglobin A1C (HbA1c) was 18.1% (35/193), fasting plasma glucose (FPG) was 21.8% (42/193), 2-h postprandial glucose (2 h PG) was 35.2% (68/193), and body mass index (BMI) was 45.6% (88/193) (See Table 1).

The analysis of sleep quality among different types of patients

High prevalence of poor sleep quality was reported, and the rate of poor sleep quality was 54.4% (105/193). The average of PSQI general score was 8.34 ± 3.98. Concerning physical pain, the t-test results indicated differences in the PSQI general score (P < 0.05), while the Mann–Whitney U test results revealed differences in various PSQI components, including subjective sleep quality, sleep duration, habitual sleep efficiency, and sleep disturbances (P < 0.05). Sleep quality among the group of pain was worse than that in no pain group. There were no statistically significant differences found in other variables (See Table 2).

The analysis of influence factors related to sleep quality

In this study, the patients were divided into poor sleep quality group (PSQI ≥ 8) and good sleep quality group (PSQI < 8). According to the PSQI scores, there were 105 for the poor sleep quality group and 88 for the good sleep quality group. The results of univariate logistic regression analysis of influence factors for sleep quality showed that there were statistically significant differences in marital status, smoking, frequency of exercise, frequency of physical examination, duration of illness, operation, dietary status, psychological burden and physical pain (P < 0.05). However, there were no significant differences in age, BMI, FPG, 2 h PG, HbA1c, drinking, life pressure, meal regularity and hypertension between the two groups (P > 0.05) (See Table 3).

Table 1 General profile of the participants (n = 193).

Category	n (%)		Category	n (%)	
Gender	male	116 (60.1)		Operation	yes	119 (61.7)	
female	77 (39.9)		no	74 (38.3)	
Smoking	yes	77 (39.9)		Hypertension	yes	86 (44.6)	
no	116 (60.1)		no	107 (55.4)	
Drinking	yes	71 (36.8)		Physical pain	yes	107 (55.4)	
no	122 (63.2)		no	86 (44.6)	
FPG (mmol/l)	<7.0	42 (21.8)		2 h PG (mmol/l)	<10.0	68 (35.2)	
≥ 7.0	151 (78.2)		≥ 10.0	125 (64.8)	
HbA1c (%)	<7.0	35 (18.1)		BMI (kg/m2 )	<24.0	88 (45.6)	
≥ 7.0	158 (81.9)		≥ 24.0	105 (54.4)	
Marital Status	married	170 (88.1)		Dietary status	good	90 (46.6)	
unmarried	4 (2.1)		general	83 (43.0)	
others	19 (9.8)		bad	20 (10.4)	
Frequency of physical examination	biannual	13 (6.7)		Frequency of exercise	≥ 5 times/week	45 (23.3)	
yearly	80 (41.5)		3-4 times/week	31 (16.1)	
biennial	22 (11.4)		1-2 times/week	33 (17.1)	
never	78 (40.4)		hardly	84 (43.5)	
Duration of illness	<1 year	38 (19.7)		Meal regularity	very regular	50 (25.9)	
1-5 years	46 (23.8)		regular	81 (42.0)	
>5 years	109 (56.5)		irregular	47 (24.4)	
Psychological burden	no	45 (23.3)		very irregular	15 (7.7)	
slightly	73 (37.8)		Life pressure	never	36 (18.7)	
moderate	35 (18.2)		rarely	61 (31.6)	
high	32 (16.6)		sometimes	48 (24.9)	
severe	8 (4.1)		often	38 (19.7)	
				always	10 (5.1)	

Table 2 The differences of PSQI component scores between pain conditions.

Indices	No	Yes	Z /t	P value	
PSQI general score	7.26 ± 3.54	9.21 ± 4.10	3.535	0.001	
Subjective sleep Quality*	80.69	110.11	3.792	<0.001	
Sleep latency*	91.35	101.54	1.313	0.189	
Sleep duration*	88.24	104.04	2.040	0.041	
Habitual sleep Efficiency*	86.98	105.06	2.310	0.021	
Sleep disturbances*	85.38	106.34	3.093	0.002	
Use of sleep medications*	95.26	98.40	1.122	0.262	
Daytime dysfunction*	89.63	102.92	1.710	0.087	
Notes.

* All data were reported as R and analyzed by the Mann–Whitney U test.

Table 3 The univariate logistic regression analysis of different influencing factors.

Category	OR (95% CI)	P value	
BMI	1.00 (0.92∼1.09)	0.967	
FPG	1.02 (0.95∼1.09)	0.574	
2 h PG	0.97 (0.92∼1.02)	0.271	
HbA1c	1.02 (0.92∼1.13)	0.727	
Age (1: ≤45, 2: 46-55, 3: 56-65, 4: >65)	1.26(0.94∼1.68)	0.121	
Smoking (1: no, 2: yes)	1.87 (1.03∼3.38)	0.037	
Drinking (1: no, 2: yes)	1.23 (0.68∼2.23)	0.477	
Operation (1: no, 2: yes)	2.08 (1.15∼3.75)	0.015	
Hypertension (1: no, 2: yes)	1.69 (0.95∼3.02)	0.072	
Physical pain (1: no, 2: yes)	2.30 (1.29∼4.12)	0.005	
Dietary status (1: good, 2: general, 3: bad)	4.02 (2.37∼6.81)	<0.001	
Duration of illness (1: <1 year, 2: 1-5 years, 3: >5 years)	1.74 (1.20∼2.52)	0.003	
Meal regularity (1: very regular, 2: regular, 3: irregular, 4: very irregular)	1.15 (0.83∼1.58)	0.390	
Frequency of physical examination (1: biannual, 2: yearly, 3: biennial, 4: never)	1.43 (1.08∼1.90)	0.011	
Frequency of exercise (1: ≥ 5 times/week, 2: 3-4 times/week, 3: 1-2 times/week, 4: hardly)	1.51 (1.19∼1.93)	0.001	
Life pressure (1: never, 2: rarely, 3: sometimes, 4: often, 5: always)	1.15 (0.89∼1.47)	0.268	
Psychological burden (1: no, 2: slightly, 3: moderate, 4: high, 5: severe)	1.29 (1.01∼1.68)	0.049	
Marital (1: married, 2: unmarried, 3: others)			
Marital(2)	0.30 (0.31∼2.97)	0.306	
Marital(3)	3.41 (1.08∼10.70)	0.035	

In order to further explore the influencing factors of sleep quality in T2DM complicated with DPN, in the univariate logistic regression analysis, certain factors showed statistically significant differences. These factors were considered as independent variables, while sleep quality (PSQI < 8 = 0, PSQI ≥ 8 = 1) served as the dependent variable. The results of multivariate logistic regression revealed that several variables were associated with an increased likelihood of poor sleep quality. These variables included lower frequency of exercise (OR: 1.40 (1.04 ∼1.89)), bad dietary status (OR: 3.42 (1.86 ∼6.29)), lower frequency of physical examination (OR: 1.49 (1.01 ∼2.20)), longer duration of illness (OR: 1.78 (1.09 ∼2.92)) and smoking (OR: 2.38 (1.17 ∼4.88)) (See Table 4).

Table 4 The multivariate logistic regression analysis of different influencing factors.

Variables	B	S.E	Wald	P value	OR (95% CI)	
Smoking	0.870	0.365	5.688	0.017	2.38 (1.17∼4.88)	
Duration of illness	0.579	0.251	5.313	0.021	1.78 (1.09∼2.92)	
Dietary status	1.231	0.310	15.720	<0.001	3.42 (1.86∼6.29)	
Frequency of physical examination	0.399	0.199	4.013	0.045	1.49 (1.01∼2.20)	
Frequency of exercise	0.338	0.153	4.865	0.027	1.40 (1.04∼1.89)	

Nomogram prediction model of factors influencing sleep quality

Frequency of exercise, dietary status, frequency of physical examination, duration of illness, and smoking were used as predictors to construct a nomogram prediction model of poor sleep quality (See Fig. 1).

Figure 1 A nomogram model for predicting sleep quality.

Discussion

China has one of the highest prevalence of diabetes in the world. With the development of social economy and changes of people’s lifestyle and dietary habits, the incidence and prevalence of diabetes are constantly increasing (Liu et al., 2017). DPN is a major cause of physical disability in diabetic patients which can occur at all stages of diabetes development. Diabetic patients often experience poor sleep quality, while those with DPN may have even worse sleep quality (Lien et al., 2020). In this study, the sleep quality and influencing factors were analyzed and nomogram prediction model in patients with T2DM who suffered DPN was constructed.

The results in this survey showed that 54.4% of patients had poor sleep quality, which was higher than previous studies. Related studies have shown that the prevalence of poor sleep quality among adults in China ranges from 8.3% to 27.7% (Dong et al., 2018; Wu et al., 2020), and a cross-sectional study by Tsai et al. (2012) (whose cut-off scores was PSQI ≥ 8) reported a prevalence of poor sleep quality in T2DM patients was 34.8%, and depending on PSQI ≥ 8, Cappuccio et al. (2009) also found that 47.1% of T2DM patients were poor sleepers. The difference in the prevalence of poor sleep quality might be due to the differences in participants and dietary cultures. Besides, the participants of this study were patients with T2DM who suffered DPN. The occurrence of complications of diabetes is not conducive to ensuring sleep quality (Lien et al., 2020), which may be the reason for the higher prevalence of poor sleep quality in this study.

In addition, there were statistically significant differences in sleep quality between the group of physical pain and the no pain group, such as the general score of PQSI, subjective sleep quality, sleep duration, habitual sleep efficiency and sleep disturbances. A previous study reported that common psychiatric disorders, such as depression and anxiety, are highly associated with sleeping problems in patients with chronic physical pain (Keilani, Crevenna & Dorner, 2018). Meanwhile, the patients of this study were T2DM patients complicated with DPN, which included some patients with painful diabetic peripheral neuropathy (PDPN). The main clinical manifestations of PDPN are distal pain of bilateral symmetrical limbs, especially at night, which can affect patients’ sleep, emotion, social function and other quality of life seriously (Alleman et al., 2015).

Marital status is considered to be an influencing factor of sleep quality, and the reason might be that married patients need to invest more time and energy to maintain a harmonious marriage relationship. However, since there were only four unmarried patients among the participants in our study, the interpretation of the results may be affected due to the small sample size, and further evidence is needed. Additionally, operation and big psychological burden was harmful to maintaining good sleep quality, which was consistent with the results of Lu et al.’s (2019) study. This might be such patients are often prone to psychological and social problems, which is not conducive to patients to maintain good sleep quality. A related study has shown that people with high BMI are more likely to have poor sleep quality (Huang, Chen & Sun, 2022) while some studies have suggested that sleep quality is not influenced by BMI. Our study showed that BMI was not an influencing factor for sleep quality, and the reason might be the uneven distribution of BMI among the patients, with the majority of them having a high BMI and only 45.6% of patients meeting the standard. Chen et al. (2020) found that FPG, 2 h PG and HbA1c are influencing factors of sleep quality in diabetic patients, while in our study, neither of them was influencing factor in sleep quality, and the reason might be that the participants included in the research were all from hospitals, and their diabetes status were less stable.

The results of multivariate logistic regression analysis displayed that the influence factors of sleep quality consisted of lower frequency of exercise, poor dietary status, lower frequency of physical examination, longer duration of illness and smoking. A nomogram prediction model based on the above independent risk factors was established to quantify and visualize the results. The model has shown that patients who hardly exercise have a high probability of poor sleep quality. Previous studies have shown that appropriate physical exercise can promote the secretion of endorphins in the human brain, which is conducive to the calmness of individual emotions (Dolezal et al., 2017; Kelley & Kelley, 2017). Results of a randomized controlled trial showed that moderate-intensity exercise training has a positive effect on sleep quality (Tseng et al., 2020). For diabetic patients, proper exercise can not only improve the ability of body tissue to combine with insulin, also obtain social support and reduce the psychological burden brought by the disease. Our study also found that the lack of regular physical examination was a risk factor for sleep quality. And the reason might be that people who have regular physical examination can better understand their own health status and promote better lifestyle habits. And regular physical examination for diabetics will enable them to intervene earlier for their existing diseases, which could reduce the diabetic status. Additionally, bad dietary status is a risk factor for sleep quality. A previous study has shown that good dietary status and diet planning can help patients to improve self-management, reduce diabetic status, and improve sleep status (Bao, Liu & Ye, 2018). A cross-sectional study in Japan showed that good dietary status and appetite would help maintain sleep quality in middle-aged and older adults (Yamamoto et al., 2020). But a relevant study has shown that different dietary patterns have different effects on sleep quality (Mondin et al., 2019). Further studies on the relationship between dietary patterns and sleep quality of chronic patients should be discussed in the future. Furthermore, the research also indicated that smoking and the duration of the illness could affect patients’ sleep quality. The risk of poor sleep quality in patients who smoke was 2.38 times higher when compared with non-smokers, and the reason might be that nicotine in tobacco stimulates the release of a variety of neurotransmitters. Besides, the longer the duration of diabetes, the poorer sleep quality. Kasteleyn et al. (2015) found that diabetic patients with a longer disease duration were more likely to develop complications, and the function of pancreatic β cells gradually declined, which was detrimental to the patient’s glycemic status and sleep quality.

There were several shortcomings in this study. Firstly, the data on sleep quality and some influencing factors in this study were reported by patients, which might lead to recall bias. However, the data in this study were obtained through face-to-face inquiries with the patients themselves, who all had good cognitive function and could communicate normally, to some degree, the recall bias could be reduced. Additionally, the patients of this study were all from the same hospital, so the sample representation was insufficient. But the hospital is a tertiary hospital with an adequate volume of patients. What’s more, the physical pain and dietary status in this study were self-reported by patients according to their own physical conditions, which were subjective to a certain extent. In a way, the results of this study could reveal the main factors affecting the sleep quality of patients with T2DM complicated with DPN, and provide a basis for the formulation of measures to improve the sleep quality of patients.

Conclusion

The investigation reported that patients with T2DM complicated with DPN have poor sleep quality. The factors that affect the sleep quality were lower frequency of exercise, bad dietary status, lower frequency of physical examination, longer duration of illness, and smoking. And the nomogram prediction model constructed in the study could be used to evaluate the risk of poor sleep quality in patients. What’s more, clinical workers and health policymakers should timely detect and effectively intervene in these factors to improve the sleep quality, which is important to improve the quality of life of T2DM patients complicated with DPN.

Supplemental Information

Supplemental Information 1 Questionnaire

Click here for additional data file.

Supplemental Information 2 Raw data

Click here for additional data file.

Additional Information and Declarations

Competing Interests

Author Contributions

Human Ethics

Data Availability

The authors declare there are no competing interests.

Lin Fu conceived and designed the experiments, performed the experiments, analyzed the data, prepared figures and/or tables, and approved the final draft.

Liping Zhong conceived and designed the experiments, performed the experiments, analyzed the data, prepared figures and/or tables, and approved the final draft.

Xin Liao performed the experiments, prepared figures and/or tables, and approved the final draft.

Lingrui Wang performed the experiments, prepared figures and/or tables, and approved the final draft.

Youyi Wang performed the experiments, prepared figures and/or tables, and approved the final draft.

Xiuquan Shi conceived and designed the experiments, authored or reviewed drafts of the article, and approved the final draft.

Yanna Zhou conceived and designed the experiments, performed the experiments, analyzed the data, prepared figures and/or tables, authored or reviewed drafts of the article, and approved the final draft.

The following information was supplied relating to ethical approvals (i.e., approving body and any reference numbers):

This study was reviewed by the Medical Research Ethics Committee of Affiliated Hospital of Zunyi Medical University (KLLY-2021-192).

The following information was supplied regarding data availability:

The raw measurements are available in the Supplemental File.

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
