# Peer review of "Deteriorated sleep quality and associate factors in patients with type 2 diabetes mellitus complicated with diabetic peripheral neuropathy"

_PeerJ, doi:10.7717/peerj.16789_

## Round 0.1 · original submission · Major Revisions

To improve the quality of the manuscript, I recommend to address all the issues raised by both reviewers.

**Language Note:** The review process has identified that the English language must be improved. PeerJ can provide language editing services - please contact us at copyediting@peerj.com for pricing (be sure to provide your manuscript number and title). Alternatively, you should make your own arrangements to improve the language quality and provide details in your response letter. – PeerJ Staff

Reviewer 1 ·

Basic reporting

I had a hard time understanding the groups. There was no information given on different groups. The authors mention "poor sleep quality and good sleep quality groups" but what is the sample size of these groups?

Experimental design

The study looks at sleep quality and its influencing factors in patients with type 2 diabetes mellitus who suffer from diabetic peripheral neuropathy. PSQI was used to investigate the sleep quality. The authors did not mention the reliability and validity of PSQI.

The influencing factors' reliability and validity are unknown. Not sure what the sub-components of the questionnaire mean: for example, "frequency of exercise" can be 30 mins of walking every day or 30 mins of strength training every other day. Need more information on this measure.

Validity of the findings

Poor sleep quality in people with diabetic peripheral neuropathy is already established. The researchers need to elaborate on "The influencing factors," which can be considered as contributing factors to poor sleep quality.

More information is needed on different groups, a sample size of different groups, and reliability/validity of outcome measures used.

There was no information on whether the data was normally distributed or not. Justification of using a parametric test was not found.

Additional comments

Grammatical errors throughout the manuscript.

Reviewer 2 ·

Basic reporting

1) There needs to be a much broader introduction to sleep. I suggest the authors have a look at the latest EASD/ADA consensus guidelines for an overarching summary of the importance of sleep in T2DM.

https://diabetesjournals.org/care/article/45/11/2753/147671/Management-of-Hyperglycemia-in-Type-2-Diabetes

2) Please consider these relevant publications in the introduction/discussion:

https://www.ncbi.nlm.nih.gov/pmc/articles/PMC5389492/
https://www.ncbi.nlm.nih.gov/pmc/articles/PMC5701896/
https://www.sciencedirect.com/science/article/pii/S2603924921000203

3) Line 70 -his seems particularly important in this cohort given the link between restless leg syndrome and neuropathy.

https://pubmed.ncbi.nlm.nih.gov/33772991/

Do you have any data on the prevalence of sleep disorders? If not, this needs to be recognised as a limitation

Experimental design

1) It would be helpful for the reader to stipulate what type of trial this is. Is it a standalone trial? If so, has it been registered on a clinical trials database?

2) Unless stipulated by the journal, please consider moving the sample size calculation to later on in the methods

3) Line 105 - Was BMI measured objectively or is this self-report?

4) Line 107 - What questionnaires were used to capture things like exercise and diet. Are they validated?

Validity of the findings

1) Please consider re-writing the results section and provide more details. For example. what constitutes "bad" dietary status? what does "hardly exercised" mean? what does a "greater psychological burden" constitute? anxiety? depression? diabetes distress? What classes as hypertension reported in Table 1? You need to provide sufficient details so that your trial could be repeated.

2) You mention reaching the standard for certain outcomes. But what are these standards?

3) The authors need to justify the exploratory analysis by sex and physical pain. Was this decided a priori? If so, it needs to be included in the introduction. Also, rather than reporting a whole host p-values, it would be better to perform interaction analyses in the first instance

Line 217 reiterates my point above (and should be removed). This is the problem with going straight into stratified analysis. You are hugely underpowered and can't draw any meaningful conclusions.

4) The discussion needs to put the results into context. Although you list a whole host of outcomes that may influence sleep quality, you present no evidence that these factors can improve sleep quality. For example, are there any interventions showing that modifications of these behaviours (e.g. more exercise, improved diet) can lead to improvements in sleep quality?

5) Consider removing Figure 2 as it

Additional comments

I appreciate that English may not be the authors' first language, but the article would benefit greatly from editing help from someone with full professional proficiency in English.

---

## Round 0.2 · Major Revisions

The authors have improved their work and addressed many concerns pointed out by the reviewers, however, there are still some issues, in particular from Reviewer 2, that need further consideration.

Reviewer 1 ·

Basic reporting

The revised manuscript is very clear and easy to understand. Thank you for considering the comments and submitting a revised manuscript. Below are some suggestions that still need to be addressed:
Abstract: under methods: please add the information on questionnaires that are given to subjects.

Experimental design

The authors have added the reliability and validity of PSQI. They have also elaborated on influencing factors. It would add to the credibility of the research study if the authors would start with the introduction of the influencing factors questionnaire that was developed by the authors and also tested its reliability and validity, followed by what the questionnaire is.

Validity of the findings

no comments.

Additional comments

no comments.

Reviewer 2 ·

Basic reporting

I wish to thank the authors for amending the article. However, some of my previous comments still need to be addressed. For example, the choice of wording still needs to be revised. Although defining what a 'bad' dietary status constitutes, it is not a particularly coherent definition. Moreover, have the choice of questions for dietary status/exercise been validated? The authors did not answer this query. Did you simply ask whether the participant exercised for 30 minutes on one or more occasion in the last month? Does 'hardly exercised' mean they did less than 30 minutes?

Experimental design

I still don't agree with the exploratory analysis, particularly when there is no interaction in the first instance.

Validity of the findings

As previously noted, more detail is needed to determine whether the results are valid.

Additional comments

It needs to be acknowledged that you don't have data on sleep disorders for these participants. Poor sleep quality and sleep disorders are not necessarily synonymous.

---

## Round 0.3 · accepted · Accept

The authors have now addressed all the remaining issues raised by the reviewers so that their manuscript is ready for publication.

Reviewer 1 ·

Basic reporting

I appreciate the revised version.

Experimental design

no comment

Validity of the findings

no comment

Reviewer 2 ·

Basic reporting

Thank you for the detailed response. My previous comments have now been addressed

Experimental design

N/A

Validity of the findings

N/A